# Scoring Verifiers: Evaluating Synthetic Verification for Code and Reasoning

**Aleksander Ficek, Somshubra Majumdar, Vahid Noroozi, Boris Ginsburg**
NVIDIA
Santa Clara, CA 15213, USA
{aficek,smajumdar,vnoroozi,bginsburg}@nvidia.com

## Abstract

Synthetic verification techniques such as generating test cases and reward modelling are common ways to enhance the coding capabilities of large language models (LLM) beyond predefined tests. Additionally, code verification has recently found great success as a critical component in improving reasoning capability of LLMs via reinforcement learning. In this paper, we propose an approach which can transform existing coding benchmarks into scoring and ranking datasets to evaluate the effectiveness of synthetic verifiers. We also propose multiple metrics to measure different aspects of the synthetic verifiers with the proposed benchmarks. By employing the proposed approach, we release four new benchmarks (HE-R, HE-R+, MBPP-R, and MBPP-R+), and analyzed synthetic verification methods with standard, reasoning-based, and reward-based LLMs. Our experiments show that reasoning can significantly improve test case generation and that scaling the number of test cases enhances the verification accuracy.[1]

## 1 Introduction

Large Language Models (LLMs) have demonstrated remarkable capabilities across various domains, particularly in code generation. Their advancements extend to solving algorithmic challenges in competitive programming, real-world software engineering tasks, and enhancing automated code testing. Recently, reasoning models such as DeepSeek-R1 (DeepSeek-AI, 2025) have found substantial improvements in math problem-solving and code generation by leveraging large-scale reinforcement learning (RL) and rule-based reward systems. In the context of coding capabilities, they utilized code execution on predefined test cases to generate the signal for RL, enabling the reasoning capability in an LLM. This highlights the importance of code verification with respect to the latest advances in reasoning models.

Although effective, an execution-based scoring approach faces a clear bottleneck due to the scarcity of the coding problems with predefined test cases. To address this constraint, many prior works have explored synthetically generated test cases and unit tests to automatically verify code quality and coverage (Schäfer et al., 2024; Chen et al., 2022). Additionally, some other works employ coding reward models to improve results on coding benchmarks (Zeng et al., 2025; Ma et al., 2025). In this paper, we collectively refer to these approaches as synthetic verifiers.

There are numerous benchmarks that assess various aspects of software engineering capabilities. Datasets such as HumanEval (HE) (Chen et al., 2021), Mostly Basic Programming Problems (MBPP) (Austin et al., 2021), and CodeContests (Li et al., 2022) are commonly used to evaluate the algorithmic and competitive programming skills of LLMs. Other benchmarks, including TESTEVAL (Wang et al., 2025), TestGenEval (Jain et al., 2024a), and SWT-Bench (Mündler et al., 2025), focus on assessing an LLM's ability to generate test cases for a given solution or feature. While these benchmarks are highly useful, they do not

---

[1]The benchmarks and code used in this paper are publicly available at https://huggingface.co/datasets/nvidia/Scoring-Verifiers and https://github.com/aleksficek/Scoring-Verifiers.

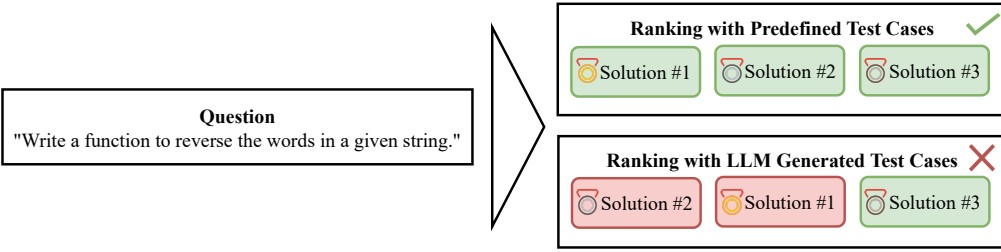

Figure 1: The figure illustrates how predefined test cases rank different solutions and how synthetic verifier rankings are compared during evaluation.

evaluate whether synthetic verification methods can effectively select better code solutions, a task we exemplify in Figure 1. Reward model benchmarks, such as RewardBench (Lambert et al., 2024), provide rankings of solutions, but these rankings are limited to preference pairs and their focus is not on coding problems. More details about past works can be found in section 6. In particular, rating test case generation suites by how well they rank solutions remains largely unexplored.

Our work is an important contribution for advancing other works that rely on high quality synthetic verifiers as a component of their training and inference pipelines. Verifiers can be used to filter synthetic code generation data (Wei et al., 2024) or to select from from several parallel generations at inference-time Zhang et al. (2025). Additionally, reinforcement learning approaches may be enhanced by fine-grained score or assessment coding solutions, more than just pass and fail, to be able to learn effectively (Liu et al., 2023a).

In this paper, we propose an approach to transform any existing coding benchmark with predefined test cases into benchmarks that asses synthetic verifiers like test case generation or reward modelling. We also propose multiple evaluation metrics to measure different aspects of the synthetic verifiers with the new benchmarks. By employing the proposed approach, we created four ranking and scoring benchmarks; HE-R, HE-R+, MBPP-R and MBPP-R+ based on the HE-R, HE-R+, MBPP-R and MBPP-R+ respectively. The new benchmarks assess how well synthetic verification methods approximate solution correctness and their ability to identify the best solution for a given problem. Then we demonstrate the benefit of these benchmarks by making quantitative observations about the ability of LLM's to generate test cases, the advantage of reasoning models in this domain, and the comparison of different synthetic verification methods like test case generation and reward models. To our knowledge, we are the first to study test case generation with reasoning models in depth. We plan to release our new benchmarks along with the code publicly.

In summary, we make the following contributions in our work:

1. We provide a recipe to transform any coding benchmark with predefined test cases into a code scoring and ranking benchmark.

2. We certify our recipe by creating code scoring and ranking versions of HumanEval and MBPP datasets: HE-R, HE-R+, MBPP-R, MBPP-R+.

3. We use our benchmark to evaluate the test case generation capability in standard and reasoning LLMs alongside coding reward models. We show that reasoning model are more effective in generating test cases compared to non-reasoning ones.

## 2 Proposed Approach

We outline our proposed process to transform a coding benchmark into a scoring and ranking benchmark in Figure 2. We assume the base benchmark contains a collections of coding questions or instructions along with their predefined test cases. We start by generating a set of solutions by employing multiple prompting techniques and filtering, with the goal of producing a set of solutions which can cover a wide spectrum of code

accuracy. This property enables us to have a better and more fine-grained evaluation of synthetic verifiers. After deduplicating the solutions, we calculate the pass rate of each solution using the predefined tests. Then we use these scores to rank the final set of solutions per question. Finally, we propose a collection of metrics to measure the quality of the synthetic verifiers on the new benchmarks. We provide more details about each stage in this section.

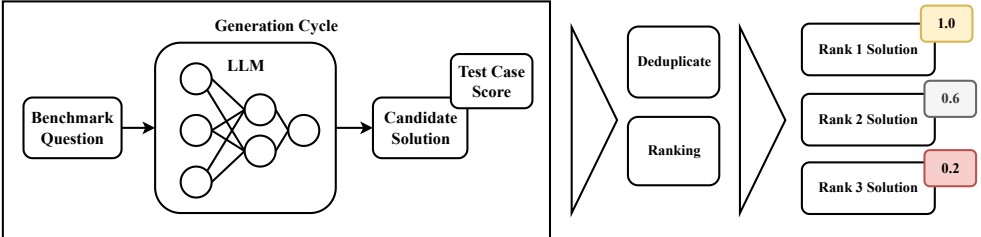

Figure 2: Diagram of the process for turning a coding dataset into a code scoring and ranking benchmark.

## 2.1 Generating Solutions

Initially, we generate some potential solutions by iterating over each question in the dataset and use multiple prompts with an LLM to produce a response/solution for the given question. This generation cycle of solutions is repeated across multiple prompts, sampling hyperparameters, and seeds to increase the diversity of solutions. We used mainly two prompts provided in Figure 8, one for generating correct and one for generating fully incorrect or partially incorrect solutions. We produce naturally and artificially incorrect responses to provide realistic failure modes while limiting self-evaluation bias. After responses are generated, the code solutions are extracted from the responses and their scores are calculated with the predefined test cases. We then aggregate all the generated solutions for each problem, building a diverse population of solutions per problem.

## 2.2 Filtering and Ranking

For each problem, we deduplicate solutions that achieve the same score (fraction of test cases passed) and tie-break using the lower average execution time to select the more optimal solutions. We always select the ground truth solution as the solutions which passes all predefined test cases. We also filter out solutions that fail completely due to non-assertion errors. This ensures we exclude solutions that may be almost correct but achieve a score of zero due to syntax errors or other trivial issues. Finally, we apply a selection algorithm to select the $k$ solutions that are most evenly distributed in terms of the fraction of test cases passed. Formally, let

$$S = \{s_1, s_2, \ldots, s_n\},$$

denote the set of deduplicated solutions, sorted in descending order such that $s_1 \geq s_2 \geq \cdots \geq s_n$, where $s_i$ represents the fraction of test cases passed by solution $i$. We assume that $s_1 = M = 1.0$ and define the minimum score $m$ as

$$m = \begin{cases} \min\{s \in S : 0 < s < 0.1\} & \text{if such } s \text{ exists} \\ s_n & \text{otherwise} \end{cases}$$

To account for cases when $k$ exceeds $n$, we define the effective selection count as:

$$k' = \min\{n, k\}.$$

For $i = 1, 2, \ldots, k' - 1$, we compute target scores:

$$T_i = 1 - \frac{i}{k'}(1 - m).$$

For each $T_i$, we select an unchosen solution that minimizes the absolute difference to $T_i$:

$$s_i^* = \underset{s \in S\{s_1^*,\ldots,s_{i-1}^*\}}{\text{argmin}} |s - T_i|.$$

Finally, we include the solution corresponding to $m$ as $s_{k'}^*$, yielding the selected set:

$$S^* = \{s_1^*, s_2^*, \ldots, s_{k'}^*\}.$$

As an example, when $m = 0.0$, our selection algorithm chooses solutions that best approximate the quantiles $(0.0, 0.25, 0.5, 0.75, 1.0)$. We continue generating solutions as described in subsection 2.1 and apply filtering stages until we achieve the desired number of uniquely scored solutions per problem. Any problem that does not reach the target $k$ after multiple rounds may either be discarded or supplemented with manually created solutions.

## 3 Proposed Benchmarks: HE-R, HE-R+, MBPP-R, MBPP-R+

### 3.1 Creation of the Benchmarks

We created four new benchmarks (HE-R, HE-R+, MBPP-R, and MBPP-R+) using the proposed approach introduced in section 2 based on the commonly used coding benchmarks of HE, HE+, MBPP, and MBPP+. We used GPT-4o (2024-11-20) OpenAI (2024a) as the generator LLM to produce the candidate solutions. The extended versions of HumanEval and MBPP include significantly more test cases, making the passing scores a more reliable proxy for the overall solution correctness. Table 1 shows statistics of the created benchmarks including dataset size, average number of test cases per problem, total number of synthetic solutions, and the average score of selected solutions. HE-R and MBPP-R have significantly lower number of test cases which can affect the quality of the code verification. In cases where the benchmark has a limited number of predefined test cases but a ground-truth solution, we recommend to follow previous methods to generate additional ground truth test cases, similar to how HE+ and MBPP+ benchmarks are created Liu et al. (2023b). In future works we recommend using a family of models at the generation stage to dilute the self-evaluation bias from an individual model.

For our transformed benchmarks, we set $k = 5$ to ensure that HE-R+ and MBPP-R+ contain at least five uniquely scored solutions per problem. For some problems, the automated process could not find enough solutions with the desired requirements, therefore we manually annotated 10 solutions for HE-R+ and 15 solutions for MBPP-R+. We use a $k$ of 2 to 5 for samples in the base versions of HumanEval (HE-R) and MBPP (MBPP-R) because of the limited number of predefined test cases. In Table 3, we perform a saturation analysis of HE+ and MBPP+ to better understand how many test cases are necessary to determine if a proposed benchmark is suitable for transformation. The lower bound of the confidence interval exceeds the conventional high correlation threshold ($\rho \geq 0.90$) at $k = 6$ for HE+ and $k = 5$ for MBPP+. These results show that a modest number of tests already yields stable rankings, consistent with our success in using the limited number of test cases available when transforming the base versions of HumanEval and MBPP.

| | HE-R | HE-R+ | MBPP-R | MBPP-R+ |
|---|---|---|---|---|
| Number of problems | 164 | 164 | 974 | 378 |
| Average number of test cases | 9.6 | 764.1 | 3.0 | 108.5 |
| Number of synthetic solutions | 742 | 820 | 3249 | 820 |
| Average score of solutions | 0.52 | 0.50 | 0.50 | 0.49 |

Table 1: Original and transformed benchmark metrics.

### 3.2 Analysis of the Test Scores

We showed the distribution of the test score differences per problem for HE-R+ and MBPP-R+ in Figure 3. Test score differences are calculated as the difference between the highest

and lowest scoring solutions for each problem. This difference can be an indicator of the correctness coverage of the generated solutions. As it can be seen, all solutions exhibit a minimum score difference of 0.5, with the majority having a difference between 0.9 and 1.0. A higher difference between the solutions shows that each solution varies in quality such that it is distinguishable by generated test cases and coding reward models.

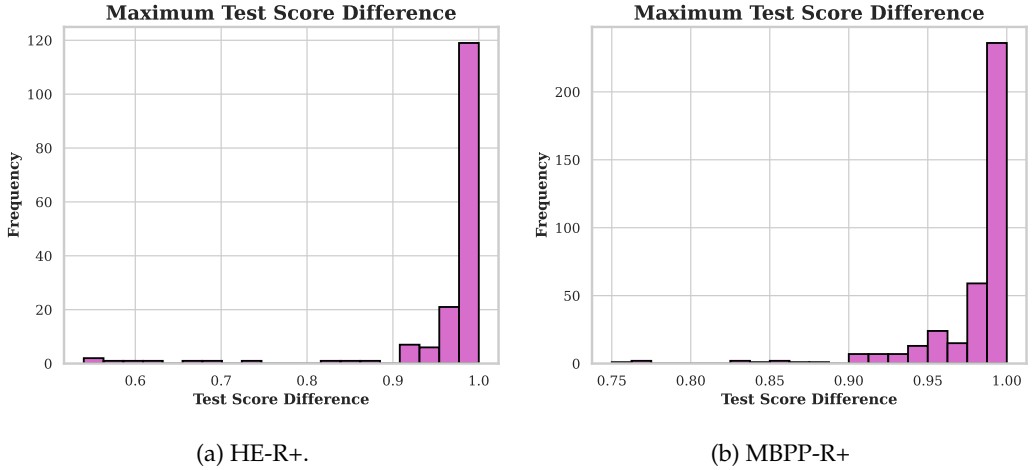

(a) HE-R+.           (b) MBPP-R+

Figure 3: Histograms of maximum difference between test case scores for each problem.

Figure 4 presents the distribution of the fraction of the tests passed for all solutions in HE-R+ and MBPP-R+. The histograms reveal a bimodal distribution, which aligns with expectations, as the ground truth solution is always included, and there is commonly a solution that fails most tests. The remaining scores conform to the typical target quantiles of $(0.0, 0.25, 0.5, 0.75, 1.0)$.

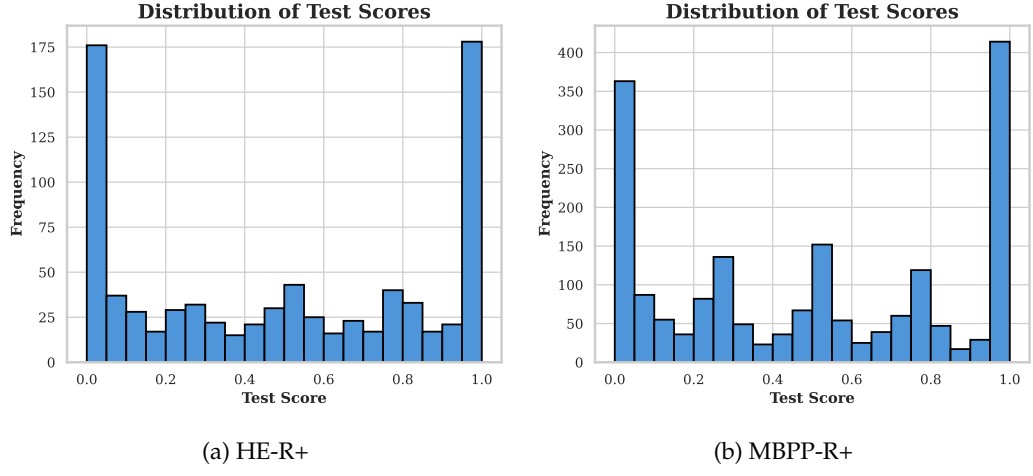

(a) HE-R+           (b) MBPP-R+

Figure 4: Histograms of distribution of test case scores in each dataset.

Histograms of the number of solutions per problem, fraction of tests passed and test score differences for all four benchmarks can be found in Appendix C. As it can be seen, HE-R and MBPP-R show similar pattern to HE-R+ and MBPP-R+ except that there are less unique candidate solution scores due to the limited number of test cases provided to differentiate their quality as shown in Table 1.

# 4 Experimental Settings

After creating the benchmarks, we used them to explore and evaluate two categories of synthetic verification methods: 1) synthetic test case generation, and 2) code reward modelling. We evaluated several models from different providers on our benchmarks (AI, 2024; Qwen et al., 2025; Hui et al., 2024; OpenAI, 2024a;b; DeepSeek-AI, 2025).

## 4.1 Test Case Generation

For evaluating models on the task of test case generation, we select a well-suited prompt (Figure 14) to generate an appropriate number of test cases for each problem in the benchmark. In our prompt, we provide two examples in HumanEval format and ensure the model wraps each test case appropriately. We used nucleus sampling with temperature of 1.0 for all the generations. Considering the large number of experiments and the limited context size of the LLMs, we generated 10 test cases per problem for our primary results as we feel this provides a reasonable coverage of edge cases. In order to compute the test scores, we executed each test case and the provided solution with a timeout of 3 seconds after noticing negligible differences in non-assertion timeout errors at higher amounts.

## 4.2 Code Reward Modelling

For evaluating reward models, we used the reward model to estimate the correctness and quality of all the solutions in the benchmarks with the prompt shown in Figure 15. We found applying a brief preamble in the prompt improves the accuracy of the reward models. We computed the reward score for each solution and normalized it using the highest and lowest scores for each problem. Then, the ranking achieved by these scores are evaluated with the different metrics. For the model Nemotron4-340B-Reward(Nvidia, 2024) we use only correctness attribute for the reward score as this fits best to the goals of our evaluations.

## 4.3 Metrics

For all the experiments, we propose to use the following metrics to compare and evaluate different LLMs. These evaluation metrics quantify a synthetic verifier's ability to correctly score and rank solution correctness.

- *Top-1 Accuracy*: Determines if the reward or the unit tests generated by an LLM can correctly rank the best solution first.

- *Spearman's $\rho$ Coefficient*: Evaluates the strength and correlation between the ranking achieved by the LLM's assessment versus the expected ranking. Expected ranking is determined by the ranking achieved by executing the predefined test cases.

- *Bottom-1 Accuracy*: Determines if the reward model or the test cases generated by the LLM can correctly rank the worst solution last.

- *Mean Absolute Error (MAE)*: Quantifies the absolute difference between the expected and estimated fraction of test cases passed.

We underscore Top-1 Accuracy and Spearman's $\rho$ coefficient as the primary metrics that encapsulate the ability for synthetic verifiers to select the best solution and rank all solutions appropriately. Bottom-1 and Mean Absolute Error are suitable secondary metrics that provide additional signals on scoring incorrect solutions alongside the delta from expected score. All ranking-based metrics are averaged across all questions within each benchmark. If test case scores result in ties, we compute the fraction of correctly ranked top solutions relative to the number of tied entries for Top-1 and Bottom-1 evaluations. For Spearman's $\rho$ Coefficient we assign the tied ranks their average position.

## 5 Results

In Table 2, we present our main results on HE-R+ and MBPP-R+ with 17 standard, reward, and reasoning-based LLMs while Table 4 presents the results on HE-R and MBPP-R.

### 5.1 Results on Test Case Generation

We find that the performance of self-generated test cases on our benchmarks generally correlates with the generating model's performance on the original HumanEval and MBPP. As it can be seen, top performing regular models on HumanEval and MBPP such as Qwen2.5-Coder-32B-Instruct, Qwen2.5-72B-Instruct, and GPT-4o perform the best among regular models. Also, larger and stronger models in each family of models outperform their smaller variants on almost all benchmarks and metrics.

| | HE-R+ | | | | MBPP-R+ | | | |
|---|---|---|---|---|---|---|---|---|
| | Top-1 | Spearman | Bottom-1 | MAE | Top-1 | Spearman | Bottom-1 | MAE |
| **Standard Models** | | | | | | | | |
| Meta-Llama-3.1-8B-Instruct | 55.9 | 0.58 | 60.4 | 0.28 | 48.5 | 0.45 | 51.1 | 0.31 |
| Meta-Llama-3.1-70B-Instruct | 66.8 | 0.67 | 71.2 | 0.24 | 61.0 | 0.63 | 67.3 | 0.25 |
| Meta-Llama-3.3-70B-Instruct | 73.8 | 0.77 | 79.7 | 0.22 | 67.7 | 0.67 | 68.7 | 0.24 |
| Qwen2.5-7B-Instruct | 71.9 | 0.76 | 73.2 | 0.23 | 58.8 | 0.64 | 68.2 | 0.25 |
| Qwen2.5-32B-Instruct | 74.9 | 0.79 | 77.5 | 0.22 | 68.8 | 0.72 | **75.0** | 0.23 |
| Qwen2.5-72B-Instruct | 78.3 | 0.80 | 76.6 | **0.21** | **71.4** | **0.73** | **75.0** | **0.22** |
| Qwen2.5-Coder-7B-Instruct | 71.2 | 0.75 | 73.8 | 0.23 | 60.1 | 0.63 | 68.3 | 0.26 |
| Qwen2.5-Coder-32B-Instruct | **79.1** | **0.83** | **80.7** | **0.21** | 68.5 | 0.72 | 73.9 | 0.23 |
| GPT-4o (2024-11-20) | 76.8 | 0.81 | 76.4 | 0.21 | 70.8 | 0.71 | 71.9 | **0.22** |
| **Reward Models** | | | | | | | | |
| AceCodeRM-7B | 68.3 | 0.65 | **62.8** | **0.23** | 70.9 | 0.52 | 40.5 | 0.27 |
| AceCodeRM-32B | **77.4** | **0.68** | 53.5 | **0.23** | 74.9 | 0.57 | 39.2 | 0.26 |
| Nemotron-70B-Reward | 60.4 | 0.61 | 53.7 | 0.24 | 69.6 | 0.53 | 39.4 | 0.27 |
| Nemotron4-340B-Reward | 76.2 | 0.67 | 59.2 | **0.23** | **75.1** | **0.59** | **46.0** | **0.25** |
| **Reasoning Models** | | | | | | | | |
| DeepSeek-R1-Distill-Qwen-32B | 78.2 | 0.78 | 74.1 | 0.22 | 70.1 | 0.65 | 68.5 | 0.24 |
| DeepSeek-R1 | 83.8 | **0.85** | 81.4 | 0.20 | 77.5 | 0.74 | 75.7 | 0.21 |
| o1-mini (2024-09-12) | 82.5 | 0.83 | 79.7 | 0.20 | 74.5 | 0.72 | 73.5 | 0.21 |
| o3-mini (2025-01-31) | **88.2** | **0.85** | **84.0** | **0.18** | **79.9** | **0.78** | **80.1** | **0.20** |

Table 2: All model results on HE-R+ and MBPP-R+.

Our findings highlight the effectiveness of test case generation once a model surpasses a certain capability threshold. Achieving 79.1% and 71.4% accuracy in differentiating the best solution with only 10 test cases is a significant challenge, requiring deep problem understanding and the ability to construct a minimal yet highly effective test suite that exposes subtle errors. Spearman's $\rho$ Coefficient and Bottom-1 values demonstrate the generated test cases also label imperfect solutions accurately. Models with at least 32B parameters demonstrate these capabilities, accurately selecting the best Top-1 and Bottom-1 solutions. However, this is not their upper limit, we show in the following subsection 5.3 that increasing the number of test cases improves performance further. Notably, these models have not been explicitly trained for test case generation yet still perform well, demonstrating significant potential for refining LLMs for this task. Figure 5a shows the breakdown of the total number of generated test cases that are passed or failed due to assertion errors and non-assertion errors on MBPP-R+, generated by DeepSeek-R1-Distill-Qwen-32B. Figure 5b similarly shows the distribution of test case scores produced by the model which exhibits a similar bimodal distribution as the ground truth test scores in Figure 4b.

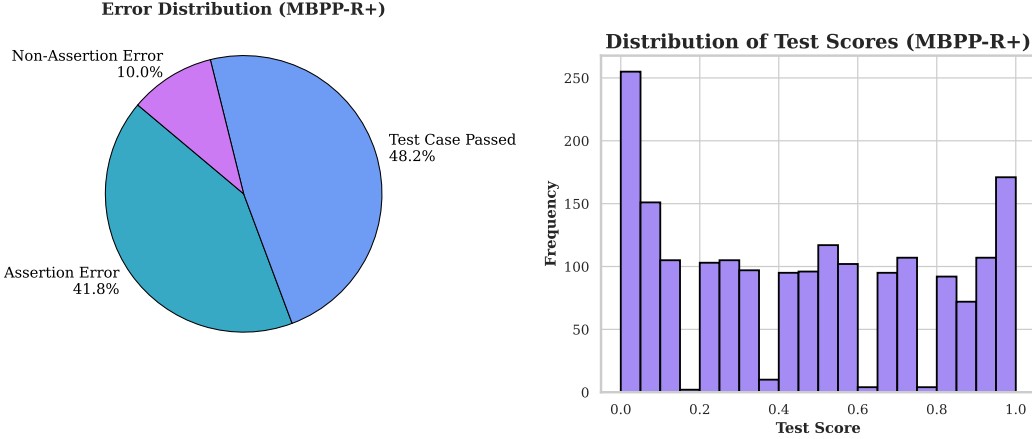

(a) Error distribution of generated test cases.     (b) Distribution of generated test case scores.

Figure 5: Analysis of test cases generated by DeepSeek-R1-Distill-Qwen-32B on MBPP-R+

## 5.2 Reasoning Model Results

We observe that the enhanced coding capabilities in reasoning models translates to improved test case generation. In a head-to-head comparison, DeepSeek-R1-Distill-Qwen-32B outperforms Qwen2.5-32B-Instruct in Top-1 accuracy but falls short in Spearman's coefficient evaluations. However, incorporating DeepSeek-R1, o1-mini, and o3-mini leads to significant improvements across all metrics, positioning them as the most effective synthetic verifiers currently available, especially when scaling the number of test cases. Figure 18 presents a sample CoT from DeepSeek-R1-Distill-Qwen-32B which illustrates how the model convincingly explores many pathways to cover potential solutions with its test cases.

## 5.3 Number of Test Cases Study

Figure 6 shows that while 10 test cases demonstrate reasonable effectiveness but scaling the test cases allows the model to better cover all of the possible cases in a problem. We see that the reasoning capabilities of DeepSeek-R1 allow the model to scale the number of test cases effectively achieving a HE-R+ Top-1 of 91.6% while plateuing on MBPP-R+. Qwen2.5-Coder-32B-Instruct and DeepSeek-R1-Distill-Qwen-32B alternatively start to plateau at the 20 test case mark. This exemplifies similar findings from Ma et al. (2025) where they scale test cases to improve reward signals in Llama3.1-70B. Further exploration is encouraged to assess the limits of reasoning and scaling of test cases to improve accuracy. Additional metrics while scaling test cases can be seen in Table 5.

## 5.4 Code Reward Model Results

Converting the original benchmarks into a scoring and ranking format enables a unique comparison of different synthetic verification methods like test case generation and reward models. From our results in Table 2, the best reward models are AceCoderRM-32B for HE-R+ and Nemotron4-340B-Reward for MBPP-R+. We find that the best performing reasoning and standard models outperform the reward models in most metrics. In similarly sized models like Qwen2.5-Coder-32B-Instruct, the Top-1 scores are competitive in HE-R+ and better in the case of MBPP-R+ while struggling in ranking the varying qualities of incorrect solutions. This could be due to subjectivity in ranking incorrect solutions, they may be functionally incorrect but qualitatively exhibiting meaningful quality. We encourage self-generated test

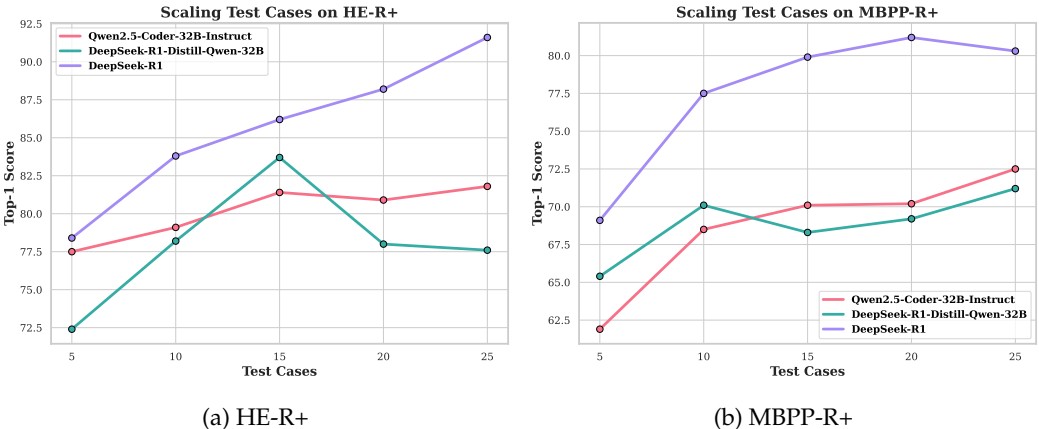

(a) HE-R+          (b) MBPP-R+

Figure 6: Top-1 scores of Qwen2.5-Coder-32B-Instruct, DeepSeek-R1-Distill-Qwen-32B and DeepSeek-R1 when scaling number of test cases.

cases as a suitable synthetic verifier for determining the correctness of a solution but see promising opportunities to further enhance reward models for coding.

## 5.5 Solution Inclusion

Finally, we examine the impact of prompting with and without a provided solution, as shown in Figure 7. All models exhibit significant performance degradation when given a potentially incorrect solution and tasked with writing test cases to evaluate it. We find that the models have a bias towards adhering to any solutions provided even when specifically prompting against this. This is supported by previous works that find LLM's to be worse at providing test cases when provided incorrect compared to correct code (Huang et al., 2024). We therefore refrain from including the solution for the rest of our experiments as a means to improve test case quality and limit self-evaluation bias.

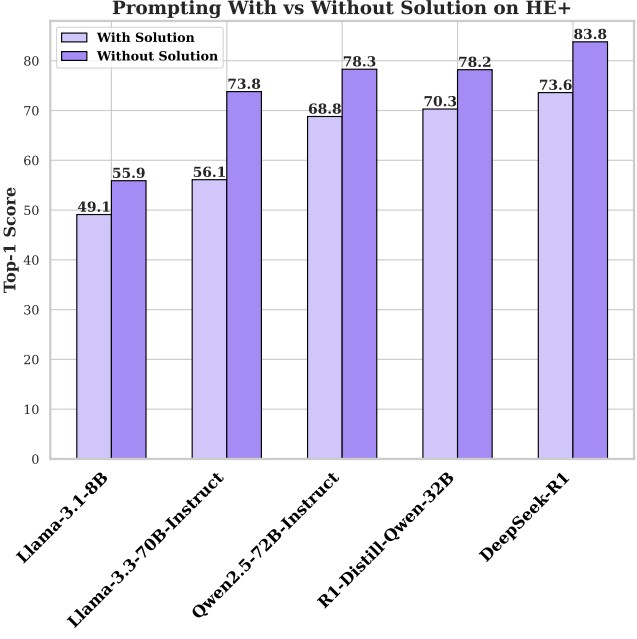

Figure 7: Generating test cases with and without solutions for various models.

## 6 Related Works

Prior work primarily validates self-generated test cases within isolated systems or limited studies. Wei et al. (2024) conducts an ablation study showing that filtering LLM-generated solutions with self-generated test cases improves synthetic data quality, evidenced by downstream supervised fine-tuning results. Light et al. (2024) compares self-generated validation tests to ground truth tests on HumanEval to highlight the impact of accurate test cases on their Scattered Forest Search method. Zhang et al. (2023) justifies its oracle verifier strategy by comparing its test cases on correct solutions. Additional techniques use test case generation to improve results on coding tasks (Liu et al., 2024; Ridnik et al., 2024; Xu et al., 2024; Ye et al., 2025). This paper unifies these approaches by introducing a benchmark for systematically assessing synthetic verifier's abilities at determining correct solutions.

As mentioned in the introduction, creating evaluations for test case generation is a well explored area. This includes many benchmarks and systems that compete over quantifying coverage, mutation testing, validity and efficiency (Wang et al., 2025; Jain et al., 2024a; Mündler et al., 2025; Jain et al., 2024b; Taherkhani & Hemmati, 2024; Ahmed et al., 2024; Peng et al., 2025; Ryan et al., 2024; Li & Yuan, 2024; Zhang et al., 2024). Crucially, we do not assess an LLM's ability to generate test cases but rather the effectivness of LLM generated test cases to determine solution quality and rank. This aligns with CodeJudge-Eval Zhao et al. (2024), which employs a similar methodology to benchmark LLM-as-a-Judge.

Our work aligns closely with reward model evaluation such as in the case of RewardBench (Lambert et al., 2024). Similarly, Zeng et al. (2025) leverages synthetic test cases to train coding reward models, evaluating them via best-of-N sampling. Ma et al. (2025) explores using generated test cases as binary signals to train a test-generating reward model, assessed through best-of-N performance. Despite these advances, a standardized benchmark for comparative evaluation remains lacking. Our work addresses this gap while also advancing test case generation across state-of-the-art standard, reasoning, and reward models.

## 7 Conclusion

We introduce a systematic approach to transform any coding benchmark with predefined test cases into a ranking and scoring benchmark for evaluating synthetic verification methods. Our method involves generating a diverse set of LLM-produced solutions, scoring them based on the fraction of test cases passed, and applying a structured filtering process to create reliably ranked datasets. We validate this approach by developing HE-R, HE-R+, MBPP-R, and MBPP-R+, which provide a standardized framework for assessing the effectiveness of synthetic verification strategies. We then use our transformed datasets to explore and uncover the effectiveness of standard, reward and reasoning based LLM's. Using our transformed datasets, we investigate the effectiveness of test case-based verification, the impact of reasoning models, and the relative strengths of reward models. Our findings reveal key insights into the performance of various LLM paradigms, highlighting the potential of reasoning-enhanced models and scaling test case generation for improved accuracy.

## Limitations

We highlight the following limitations in our work. First, the verification benchmarks produced using our approach rely directly on the ground truth of the original benchmark for benchmark transformation. Therefore, the produced verifier benchmark quality is directly tied to the reliability and accuracy of the original benchmark. Secondly, we were practically constrained by the output context length pricing when generating test cases using the API-based reasoning models. This led us to strategically selecting 10 test cases based on the plateau of performance from non-reasoning models at that value while remaining within this the pricing constraint.

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

# A Producing Solution Prompt

---

**Producing Correct Solutions Prompt**

Using only Python code, write a solution to the given coding problem.
Here are other guidelines for completing this task:

1. Enclose the code in a python code block "'python.
2. Do not include any unit tests in your answer, only generate the function.
3. The code must still compile, the only errors in the code should be logical.
4. Include any necessary imports with your code, only import libraries included in the standard library.

Question:
{question}

Answer:

---

**Producing Partially Correct Solutions Prompt**

Using only Python code, write a somewhat incorrect solution to the given coding problem.
Do not provide any hints as to what is the mistake. Here are other guidelines for completing this task:

1. Enclose the code in a python code block "'python.
2. Do not include any unit tests in your answer, only generate the function.
3. The code must still compile, the only errors in the code should be logical.
4. Include any necessary imports with your code, only import libraries included in the standard library.
5. Do not add any hints as to the error you made.

Here are some suggestions:
- Do not handle negative numbers
- Do not handle duplicate values
- Introduce rounding errors
- Ignore the last element in a list
- Only handle specific values
- Only works for certain ranges of values or lengths

Question:
{question}

Answer:

---

Figure 8: Prompt templates for producing solutions

## B  Saturation Analysis

| $k$ | $\rho_{\text{mean}}$ | $\rho_{95\%\,\text{low}}$ | $\rho_{95\%\,\text{high}}$ | $\sigma_\rho$ |
|---|---|---|---|---|
| 1 | 0.725 | 0.722 | 0.728 | 0.016 |
| 2 | 0.815 | 0.813 | 0.817 | 0.011 |
| 3 | 0.857 | 0.855 | 0.859 | 0.009 |
| 4 | 0.879 | 0.878 | 0.881 | 0.009 |
| 5 | 0.895 | 0.894 | 0.897 | 0.008 |
| 6 | 0.905 | 0.903 | 0.906 | 0.008 |
| 7 | 0.913 | 0.912 | 0.914 | 0.006 |
| 8 | 0.919 | 0.917 | 0.920 | 0.007 |
| 9 | 0.925 | 0.924 | 0.926 | 0.007 |
| 10 | 0.929 | 0.927 | 0.930 | 0.006 |

(a) HE+

| $k$ | $\rho_{\text{mean}}$ | $\rho_{95\%\,\text{low}}$ | $\rho_{95\%\,\text{high}}$ | $\sigma_\rho$ |
|---|---|---|---|---|
| 1 | 0.739 | 0.737 | 0.740 | 0.008 |
| 2 | 0.834 | 0.833 | 0.836 | 0.007 |
| 3 | 0.875 | 0.874 | 0.876 | 0.005 |
| 4 | 0.900 | 0.899 | 0.901 | 0.006 |
| 5 | 0.915 | 0.914 | 0.916 | 0.004 |
| 6 | 0.925 | 0.924 | 0.926 | 0.004 |
| 7 | 0.933 | 0.932 | 0.933 | 0.004 |
| 8 | 0.938 | 0.938 | 0.939 | 0.004 |
| 9 | 0.943 | 0.942 | 0.944 | 0.004 |
| 10 | 0.948 | 0.948 | 0.949 | 0.003 |

(b) MBPP+

Table 3: Saturation analysis of number of test cases with HE+ and MBPP+ benchmarks.

## C  Benchmark Analysis Visualizations

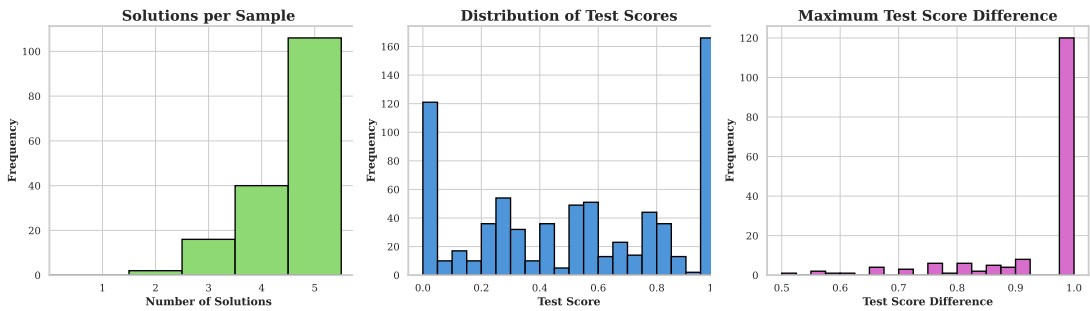

Figure 9: HumanEval (HE-R) benchmark analysis.

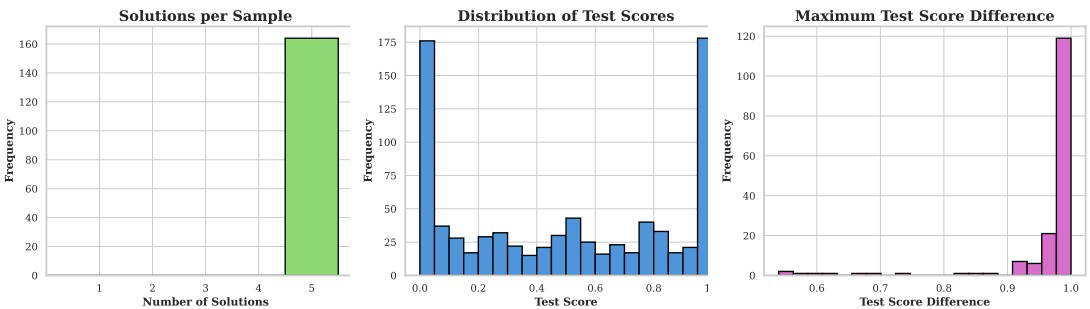

Figure 10: HumanEval Plus (HE-R+) benchmark analysis.

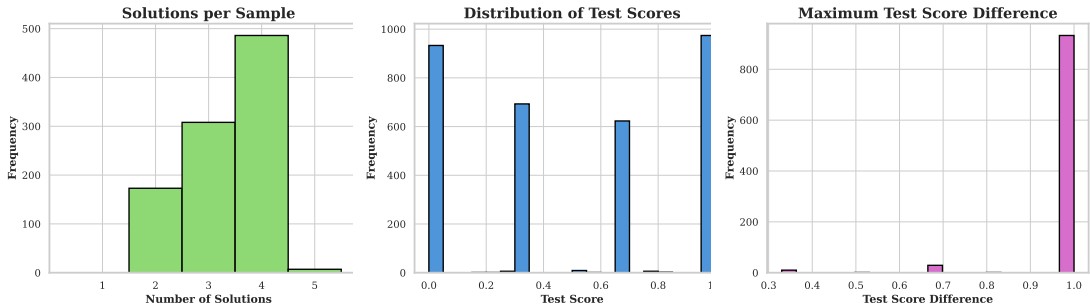

Figure 11: MBPP (MBPP-R) benchmark analysis.

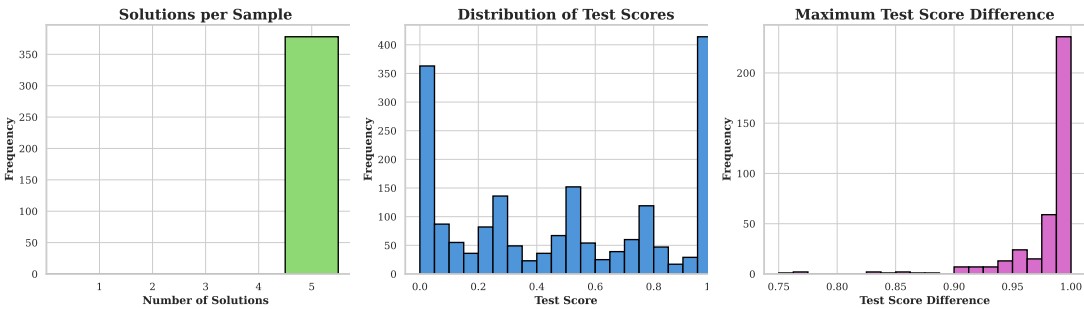

Figure 12: MBPP Plus (MBPP-R+) benchmark analysis.

## D   Test Case Generation Prompts

---

**Test Case Generation Without Solution Prompt**

You are an expert at writing assertion test cases and below is a question with function signature and test cases. You must generate 10 assert test cases that will be used to evaluate the code solution's correctness. You must adhere to the provided function signature and test case format. Here are some examples that you should use as a reference:

Question:

```
from typing import Optional
def first\_repeated\_char(s: str) -> Optional[str]:
    """
    Find the first repeated character in a given string.
    >>> first\_repeated\_char("abbac")
    'a'
    """
```

Test Cases:

```
<assertion>assert first_repeated_char("!@#$%^&*!") == "!"</assertion>
<assertion>assert first_repeated_char("abcdedcba") == "d"</assertion>
<assertion>assert first_repeated_char("") == "None"</assertion>
<assertion>assert first_repeated_char("aaaa") == "a"</assertion>
<assertion>assert first_repeated_char("a") == "None"</assertion>
```

Here are guidelines for writing the assertion test cases:

1. You must wrap each assertion test case with tags ¡assertion¿ and ¡/assertion¿.
2. Do not start the assert with any indents or spaces.
3. You must not import any unit testing libraries for the assertions such as "unittest" or "pytest".
4. Each assertion must be complete and immediately executable. Assume the code solution is provided, do not repeat it.
5. Avoid unnecessary string literals, incorrect escaping, wrapping in """python" or other redundancies.
6. Remember, it is your responsibility to carefully read the question and generate test cases that will evaluate the correctness of the solution.

Here is the question you must provide assertion test cases for:

Question: {question}

Test Cases:

---

Figure 13: Prompt template for test case generation without solution

---

**Test Case Generation With Solution Prompt**

You are an expert at writing assertion test cases and below is a question with function signature and completed code solution. You must generate 10 assert statements that will be used to evaluate the code solution's correctness which may or may not be correct. Here are some examples that you should use as a reference:

Question:

```
from typing import Optional
def first\_repeated\_char(s: str) -> Optional[str]:
    """
    Find the first repeated character in a given string.
    >>> first\_repeated\_char("abbac")
    'a'
    """
```

Solution:

```
from typing import Optional
def first_repeated_char(s: str) -> Optional[str]:
    """
    Find the first repeated character in a given string.
    >>> first_repeated_char("abbac")
    'a'
    """
    for index, c in enumerate(s):
        if s[:index + 1].count(c) > 1:
            return c
    return None
```

Test Cases:

```
<assertion>assert first_repeated_char("!@#$%ˆ&∗!") == "!"</assertion>
<assertion>assert first_repeated_char("abcdedcba") == "d"</assertion>
<assertion>assert first_repeated_char("") == "None"</assertion>
<assertion>assert first_repeated_char("aaaa") == "a"</assertion>
<assertion>assert first_repeated_char("a") == "None"</assertion>
```

Here are guidelines for writing the assertion test cases:
1. You must wrap each assertion test case with tags ¡assertion¿ and ¡/assertion¿.
2. Do not start the assert with any indents or spaces.
3. You must not import any unit testing libraries for the assertions such as "unittest" or "pytest".
4. Each assertion must be complete and immediately executable. Assume the code solution is provided, do not repeat it.
5. Avoid unnecessary string literals, incorrect escaping, wrapping in """"python" or other redundancies.
6. Remember, it is your responsibility to carefully read the question and generate test cases that will evaluate the correctness of the solution.

Here is the question and code solution you must provide assertion test cases for:

Question: {question}

Solution: {solution}

Test Cases:

Figure 14: Prompt template for test case generation without solution

# E   Reward Model Prompts

| Reward Model User Prompt |
|---|
| You are given a coding problem for which you need to generate/complete a solution that is as accurate as possible.

Please complete the function with the Python programming language.

This is the problem you must solve: {question} |

| Reward Model Assistant Prompt |
|---|
| Here is the solution to the given problem: {solution} |

Figure 15: Prompt templates for reward model user and assistant turns

# F  Supplementary results

| | HE-R | | | | MBPP-R | | | |
|---|---|---|---|---|---|---|---|---|
| | Top-1 | Spearman | Bottom-1 | MAE | Top-1 | Spearman | Bottom-1 | MAE |
| **Standard Models** | | | | | | | | |
| Meta-Llama-3.1-8B-Instruct | 63.6 | 0.69 | 70.6 | 0.24 | 55.4 | 0.42 | 56.4 | 0.35 |
| Meta-Llama-3.1-70B-Instruct | 74.2 | 0.80 | 78.8 | 0.18 | 72.0 | 0.68 | 72.7 | 0.25 |
| Meta-Llama-3.3-70B-Instruct | 81.6 | 0.89 | 84.8 | 0.14 | 76.1 | 0.73 | 76.2 | 0.23 |
| Qwen2.5-7B-Instruct | 81.2 | 0.86 | 78.4 | 0.18 | 72.5 | 0.66 | 72.5 | 0.26 |
| Qwen2.5-32B-Instruct | 85.7 | 0.90 | **86.9** | **0.13** | **78.3** | **0.75** | **78.8** | **0.22** |
| Qwen2.5-72B-Instruct | 85.3 | 0.90 | 84.2 | 0.14 | 76.9 | 0.74 | 78.3 | **0.22** |
| Qwen2.5-Coder-7B-Instruct | 81.6 | 0.86 | 82.4 | 0.17 | 67.2 | 0.61 | 69.4 | 0.27 |
| Qwen2.5-Coder-32B-Instruct | **89.0** | **0.91** | 86.1 | **0.13** | 76.5 | 0.73 | 77.6 | 0.23 |
| GPT-4o (2024-11-20) | 81.8 | 0.88 | 84.6 | 0.14 | 74.9 | 0.71 | 76.3 | **0.22** |
| **Reward Models** | | | | | | | | |
| AceCodeRM-7B | 71.3 | 0.68 | 18.9 | 0.22 | 71.2 | 0.53 | 35.7 | 0.26 |
| AceCodeRM-32B | 80.5 | **0.75** | **29.3** | 0.21 | **75.8** | **0.60** | **38.6** | **0.24** |
| Nemotron-70B-Reward | 65.2 | 0.65 | 16.5 | 0.22 | 45.7 | 0.31 | 24.9 | 0.35 |
| Nemotron4-340B-Reward | **82.9** | 0.72 | 20.7 | **0.21** | 66.0 | 0.55 | 37.5 | 0.26 |
| **Reasoning Models** | | | | | | | | |
| DeepSeek-R1-Distill-Qwen-32B | 81.8 | 0.86 | 81.0 | 0.13 | 74.1 | 0.68 | 73.0 | 0.23 |
| DeepSeek-R1 | **90.9** | **0.92** | **84.0** | **0.11** | **81.2** | **0.77** | **79.2** | **0.20** |

Table 4: Model results on HE-R and MBPP-R.

| | HE-R | | | | MBPP-R | | | |
|---|---|---|---|---|---|---|---|---|
| | Top-1 | Spearman | Bottom-1 | MAE | Top-1 | Spearman | Bottom-1 | MAE |
| **Qwen2.5-Coder-32B-Instruct** | | | | | | | | |
| 5 Test Cases | 77.5 | 0.81 | 75.7 | 0.22 | 61.9 | 0.67 | 67.1 | 0.24 |
| 10 Test Cases | 79.1 | 0.83 | 80.7 | **0.21** | 68.5 | 0.72 | 73.9 | 0.23 |
| 15 Test Cases | 81.4 | **0.82** | **82.9** | **0.21** | 70.1 | 0.72 | 76.7 | 0.22 |
| 20 Test Cases | 80.9 | 0.80 | 82.2 | **0.21** | 70.2 | 0.71 | 76.7 | 0.23 |
| 25 Test Cases | **81.8** | 0.81 | 80.4 | 0.22 | **72.5** | **0.73** | **80.1** | **0.22** |
| **DeepSeek-R1-Distill-Qwen-32B** | | | | | | | | |
| 5 Test Cases | 72.4 | 0.76 | 72.2 | 0.23 | 65.4 | **0.65** | 64.1 | 0.24 |
| 10 Test Cases | 78.2 | 0.78 | 74.1 | 0.22 | 70.1 | **0.65** | 68.5 | 0.24 |
| 15 Test Cases | **83.7** | 0.77 | 74.4 | 0.21 | 68.3 | 0.62 | 63.1 | 0.24 |
| 20 Test Cases | 78.0 | **0.81** | **79.9** | **0.20** | 69.2 | 0.64 | **69.0** | **0.23** |
| 25 Test Cases | 77.6 | 0.76 | 74.1 | 0.21 | **71.2** | 0.63 | 65.5 | **0.23** |
| **DeepSeek-R1** | | | | | | | | |
| 5 Test Cases | 78.4 | 0.79 | 74.4 | 0.21 | 69.1 | 0.71 | 69.4 | 0.23 |
| 10 Test Cases | 83.8 | 0.85 | 81.4 | 0.20 | 77.5 | 0.74 | 75.7 | 0.21 |
| 15 Test Cases | 86.2 | 0.84 | 84.7 | 0.19 | 79.9 | **0.76** | **77.8** | 0.20 |
| 20 Test Cases | 88.2 | **0.86** | 85.4 | **0.18** | **81.2** | **0.76** | 76.7 | **0.19** |
| 25 Test Cases | **91.6** | **0.86** | 85.4 | 0.19 | 80.3 | 0.75 | 77.1 | 0.20 |

Table 5: Test case scaling results.

# G   Model Test Case Scoring Results

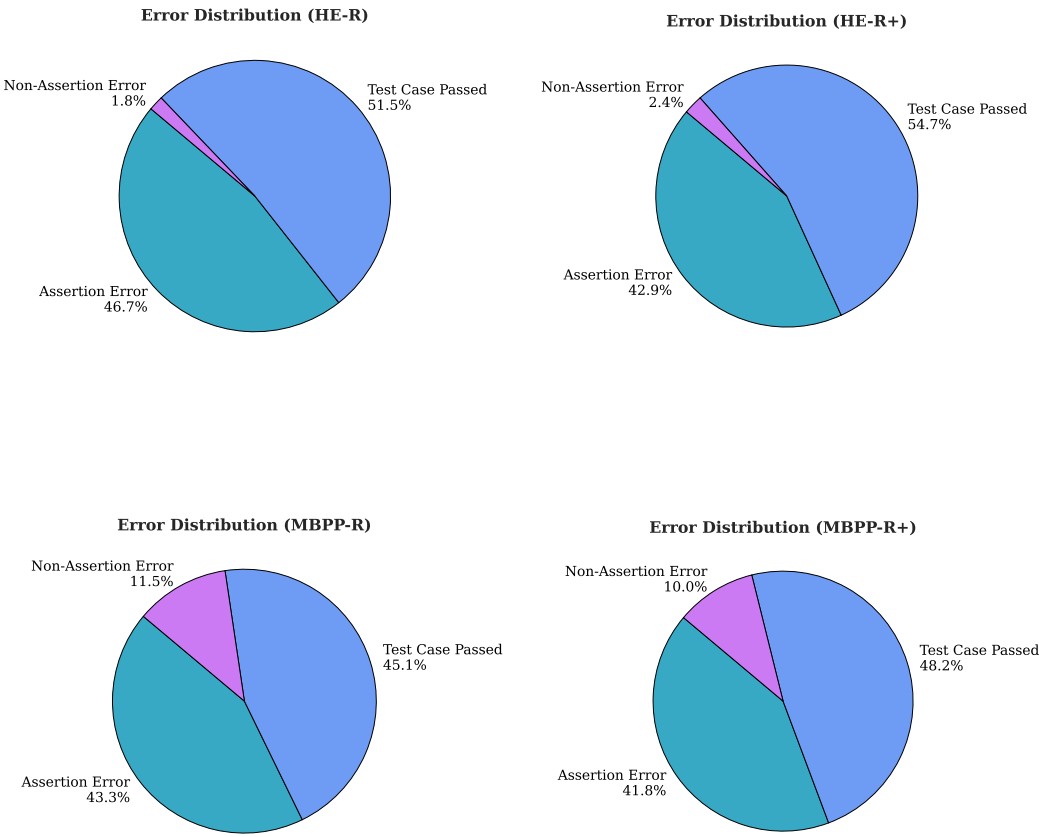

Figure 16: DeepSeek-R1-Distill-Qwen-32B test case error distributions.

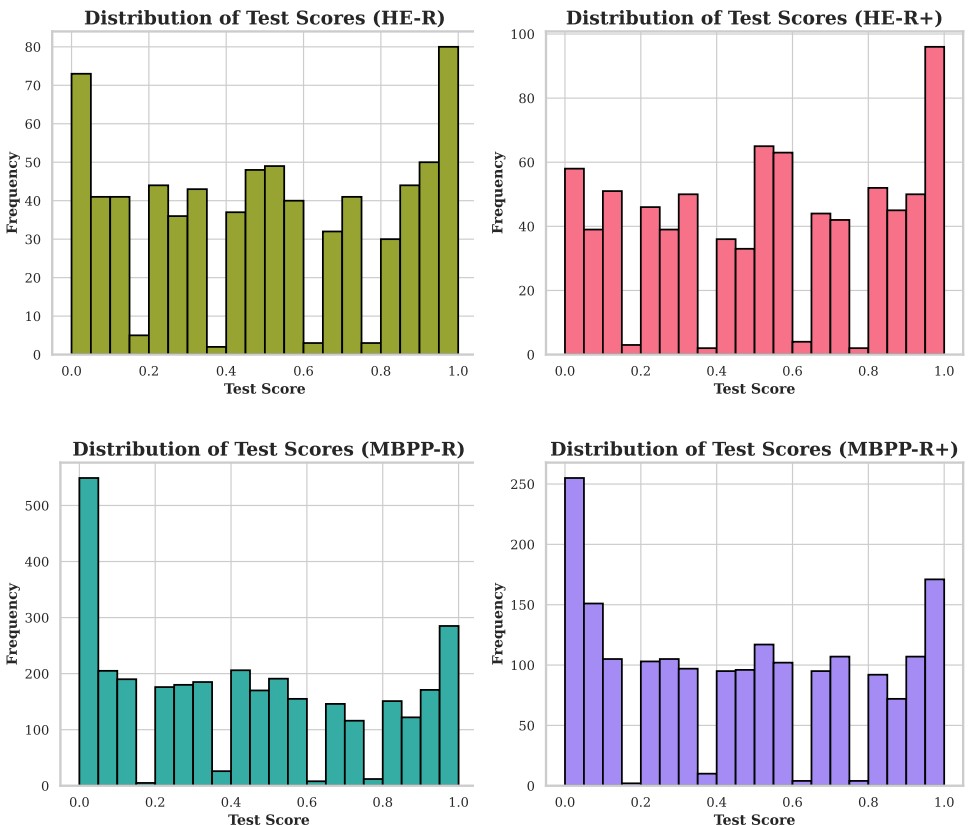

Figure 17: DeepSeek-R1-Distill-Qwen-32B test case score distributions.

# H   Reasoning Model Chain-of-Thoughts

---

**Sample MBPP Question (MBPP Question 809)**

Write a function to check if each element of second tuple is smaller than its corresponding element in the first tuple.

```
assert check_smaller((1, 2, 3), (2, 3, 4)) == False
```

---

**DeepSeek-R1-Distill-Qwen-32B Reasoning Trace**

Alright, I need to write 10 assertion test cases for the function check_smaller, which takes two tuples and returns True if each element in the second tuple is smaller than the corresponding element in the first tuple.

Let me think about different scenarios that the function might encounter.

First, the sample test case provided is when all elements of the second tuple are larger. That's a good start.

Then, I should consider cases where the second tuple has smaller elements for all positions, which should return True.

Next, it's important to test cases where the second tuple is empty. According to the function's intended behavior, since there are no elements to compare, it should return True.

Also, leading or trailing zeros might be tricky, so I should include a test case where the first tuple has a zero and the second has a positive number, ensuring it returns False.

What about negative numbers? If both tuples have negative numbers, the function should correctly compare them. For example, (-5, -6) compared to (-4, -7) should return False because the first element is smaller, but the second is larger. Wait, no the function should return True only if all second elements are smaller. In this case, the first elements of the second tuple (-4, -7) is larger, so the result is False. Wait, that might not be correct. Wait, the function should return True only if every element in the second tuple is smaller. So if the second tuple has any element not smaller, return False. So, if the first tuple is ( -5, -6 ), and the second is ( -4, -7 ), then: First element: -4 is larger than -5, not smaller, so function should return False. I'll include that case as well.

I should also consider tuples of different lengths. Oh, but according to the function signature, it's taking two tuples. What if they're of different lengths? For example, tuple1 is (1, 2) and tuple2 is (1, 2, 3). Do we need to handle that? The problem description doesn't mention it, so perhaps the function assumes that the tuples are of the same length. So I can test with same-length tuples, but also think about varying lengths. But since the function's docstring doesn't specify, maybe I can assume the tuples are of the same length. However, I'll include a test case where the second tuple is longer, but the function only checks up to the shorter length.

---

Figure 18: DeepSeek-R1-Distill-Qwen-32B reasoning trace example

