# OpenReview forum: "Scoring Verifiers: Evaluating Synthetic Verification for Code and Reasoning"
_colmweb.org/COLM/2025/Conference — COLM 2025_

### Official Review · Reviewer_49g2 · 2025-05-12

**Rating:** 7
**Confidence:** 3
**Ethics Flag:** 1

**Summary:**

This paper is in the domain of test case generation for programs and precisely on the evaluation of test case generation. It generates a large number of solutions to benchmark questions, with a voluntary varying quality. Next, it uses these solutions to evaluate test case generation models, as the generated test cases should give high scores to high quality solutions and low ones to low quality solutions.

This work also allows to compare test case generation models with another kind of synthetic verification model, namely reward models. The results show that the latter are less efficients to judge solutions quality.

**Questions To Authors:**

The justification of generating 10 test cases by a "feeling" of adequacy is not acceptable. And why a 3 seconds timout, in the same paragraph and not 2, 2.5 or 4 ?


A few typos or presentation problems:
- abstract: "a an"
- page 2: "a benchmarks"
- page 4: "Creations of the…"
- page 4: "used use"
- page 5: correct citations like "Qwen Qwen…". The repetition is not necessary
- pahe 5, line 158: "For evaluating" ?

**Reasons To Accept:**

The paper is well written. The experiments are convincing at confirming the utility of their solutions generation as a way to evaluate test case generation models.

**Reasons To Reject:**

I may have missed important flaws, but I do not see obvious reasons to reject this paper.

---

> ### Author Response · Authors · 2025-06-01
> **Addressing questions and concerns**
>
> We sincerely appreciate you taking the time to review our paper. We are glad that you enjoyed our work and recognize its importance!  We will be sure to clean up any grammatical disfluencies in the camera ready version of our work. We justify our use of 10 test cases based on the plateau of performance from non-reasoning models in Figure 6. We were practically constrained by the output context length pricing when generating test cases using the API-based reasoning models and strategically selected a number of tests appropriate for the models we evaluated within this constraint. We chose a 3-second timeout after testing larger timeouts such as 5 seconds and noticing a negligible differences in non-assertion timeout errors displayed in Figure 5.a. We are happy to include these points in the experimental settings of our paper.

---

> > ### Comment · Reviewer_49g2 · 2025-06-04
> >
> > I have read all comments and rebuttals. Some critics of refusing reviewers were justified, but the answers of authors are also good. I thus confirm my rating.

---

### Official Review · Reviewer_WWNV · 2025-05-13

**Rating:** 7
**Confidence:** 4
**Ethics Flag:** 1

**Summary:**

This paper presents an approach to scoring code generation in a nuanced way that supports comparison/ranking by leveraging RL preparation for reasoning models.  This is important work since correct vs incorrect designations are too blunt to adequately address the need for finetuning.  The prior work section is nicely written, with each paragraph that summarizes an area of work ending with a statement about how this work contributes over-and-above what came before.

**Questions To Authors:**

I don't have any major questions.  As mentioned above, the related work section does talk about why the approach described is an advance over past work, but I wonder what the authors would argue is their main spark of insight/innovation? How should this work push thinking forward in the field?

**Reasons To Accept:**

There is a clear need for this work.  The technical approach is clearly described, and the results look promising.  The graphs in the results section especially make a compelling case for the strength of the results.  New benchmark datasets for sharing are created and validated.

**Reasons To Reject:**

No real reason to reject -- this is a well done piece of work.  If I was pushed to find a weakness, I would say it's that the paper takes advantage of resources that are on the frontier currently in the field, but though the approach is resourceful, it's not particularly creative.

Smaller comment: the writing is sloppy.  I see many grammatical disfluencies throughout.

---

> ### Author Response · Authors · 2025-06-01
> **Addressing questions and concerns**
>
> We sincerely appreciate you taking the time to review our paper. We are glad that you enjoyed our work and recognize its importance! We will be sure to clean up any grammatical disfluencies in the camera ready version of our work. In response to your questions about our main spark of insight and innovation, we were motivated to create this work as we were experimenting with synthetic verifiers but were actively bearing the lack of current benchmarks that currently exist in the space. RewardBench is one of the only benchmarks to evaluate verifiers and its coding evaluation is severely limited. Our idea came as we explored ways to evaluate coding verifiers directly instead of evaluating a downstream generator model trained on a reward model’s signal. We believe this work significantly advances synthetic verifier evaluation, particularly given the increasing popularity of generative reward modeling, which is notably compute-intensive at inference time.

---

> > ### Comment · Reviewer_WWNV · 2025-06-03
> > **Thank you**
> >
> > I have read your rebuttal.

---

### Official Review · Reviewer_3nRV · 2025-05-13

**Rating:** 4
**Confidence:** 3
**Ethics Flag:** 1

**Summary:**

This paper proposes a pipeline to repurpose datasets (HE-R, HE-R+, MBPP-R, MBPP-R+) for benchmarking the performance of synthetic verifiers based on the existing code benchmarks. The experiments on the synthetic verifiers with different LLMs show that the reasoning models can significantly improve the quality and scale of test case generation

**Questions To Authors:**

What if the synthesis of incorrect or partially correct solutions diverges from the real incorrect solutions that LLMs will generate? Will this lead to a misalignment between the evaluation result of verifiers and their actual performance when used as a reward model for ranking or RL learning?

**Reasons To Accept:**

1. It provides a pipeline to transfer the existing code generation benchmark to the code verifier benchmark.
2. Based on the repurpose benchmark, it shows several interesting findings on the performance of the LLM-based verifier in test case generation and reward modeling. It also provides a detailed analysis of the error patterns and the effect of test case number.

**Reasons To Reject:**

1. There is no guarantee of the reliability of the new benchmark. The ground-truth score and rank of the repurpose benchmark are based on the limited number of predefined test cases. As shown in Table 1, the number of predefined test cases in HE-R and MBPP-R is small, and these cases may not be specific enough to identify the correct and incorrect solutions. There is no specific way mentioned in the proposed transform pipeline on how to measure the reliability of the transformed benchmark. For example, how many test cases in the original dataset can make the new benchmark reliable enough for evaluating the verifier?
2. There may exist self evaluation bias: the verifiers that share the same LLMs with the solution generation model may have a higher score. For example, it is unclear whether the better performance in GPT-4o, o1-mini, and o3-mini is affected by the fact that these incorrect solutions are generated by GPT-4o.

---

> ### Author Response · Authors · 2025-06-01
> **Addressing questions and concerns**
>
> We sincerely appreciate you taking the time to review our paper and address the concerns you mentioned in the review below.
>
> __1. Lack of ability to guarantee the reliability of the benchmark.__
>
> This critique highlights a genuine limitation of the benchmarks we produce. We selected the original MBPP and HE datasets deliberately to rigorously test the limitations of our approach compared to their enhanced counterparts. We do our best to explore this limitation by making comparisons of the number of test cases [Table 1], the difference in test case scores [Section 3.2, Figures 9 - 12] and how the final evaluations are similar despite the differences [Section 5.1]. However, since we rely directly on the ground truth of the original benchmark for benchmark transformation, our resultant benchmark’s reliability is directly tied to the reliability of the original benchmark. The promise of our work is not to guarantee benchmark reliability but to effectively convert a benchmark that evaluates code generation into one that evaluates code verification. Thus, the reviewer’s important question of "how many test cases make the new benchmark reliable enough?" parallels the same question asked to the original code generation dataset. This is a very important distinction which we will also explicitly address by adding a limitations section and convey this concept.
>
> __2. Potential self-evaluation bias.__
>
> This is another constructive comment on the paper which we attempt to address two-fold in our paper at the benchmark generation and verifier evaluation sections but do not communicate directly enough. Firstly, we employ our extensive generation and filtering strategy [Section 2.1, 2.2] and discuss its impacts in [Section 3.2] as a means to mitigate self-evaluation bias in the generator model. For example, [Figure 8] shows how we explicitly prompt for incorrect failure modes alongside ones naturally produced by the model to reduce self-evaluation bias while remaining close to real incorrect solutions. Because we then filter by selecting target levels of correctness with the provided test cases we limit bias by leveraging the diversity produced from hundreds of unique candidate generations to each problem. Any additional bias that exists in terms of style of code and readability is not relevant as our benchmark only tests for correctness. Secondly, we study the impact of self-evaluation bias in [Section 5.5] where we find that models generally produce lower quality test cases when provided the solution. Because of this, the standard and reasoning model verifiers in our main results such as GPT-4o, o1-mini and o3-mini are given only the problem and not the solution when generating test cases. Since the solution is not available to the model during evaluation inference any self-evaluation bias is heavily mitigated. Reward models inherently require the solution to operate so we intentionally selected a generator model that differs in family from the reward models to minimize self-evaluation bias. We will be sure to emphasise these elaborations in their respective sections. We will also include how self-evaluation bias from a particular model can be diluted by using a family of models at the generation stage which we believe elevates our approach for future works looking to convert generation benchmarks to verifier benchmarks.
>
> __3. Divergence between synthetic and real incorrect solutions.__
>
> The concept of misalignment between real and synthetic data is a common concern with any synthetic data approach such as our work. While this is difficult to quantify and guarantee we do our best to combat this by employing a variety of prompts when introducing synthetic “incorrectness” in addition to natural incorrect solutions [Figure 8]. Finally, our benchmark compares models on a relative basis instead of absolutely so any remaining misalignment in our benchmark would apply to all evaluated models. Empirically, we observe that downstream verifier performance on our benchmarks tracks with their relative performance on the original benchmarks, indicating minimal misalignment in practical scenarios.
>
> We are grateful for your thoughts on our work and are eager to hear any comments you may have!

---

> > ### Comment · Reviewer_3nRV · 2025-06-09
> >
> > I appreciate the authors' detailed response, which addresses some of my concerns. However, I remain concerned about the quality of the transformed verification benchmark.
> >
> > The paper's main contribution is proposing a pipeline to transform code benchmarks into ones that evaluate code verifiers. As the authors acknowledge, evaluating the reliability of this new benchmark is challenging. While the authors note that original code benchmarks face similar reliability issues, there is a crucial difference: code benchmarks provide ground-truth solutions as references, making the limited number of test cases less problematic.
> >
> > In contrast, the proposed verification benchmark lacks such a reference point. The "ground-truth" rankings may themselves be incorrect due to insufficient test cases, and determining an adequate number of test cases remains unclear. This creates a concerning scenario: a fundamentally incorrect solution (potentially generated by the Producing Correct Solutions Prompt) might pass all test cases in the original benchmark and receive a high ranking in the verification benchmark. A verifier that appropriately ranks this flawed solution as low-quality would be penalized with a poor score, despite performing correctly.
> >
> > This fundamental issue undermines the benchmark's ability to reliably evaluate verifier performance, as the evaluation metric itself may be based on flawed ground truth. There should be at least quality control checks for the reliability of this verification benchmark, or empirical conclusions on how many test cases in the original benchmark are enough for such a transformation.

---

> > > ### Author Response · Authors · 2025-06-09
> > >
> > > We thank the reviewer for the follow-up and address the remaining concern of benchmark reliability.
> > >
> > > **1. Reference solutions vs. test-case oracle**
> > >
> > > In both the original generation benchmarks and our transformed verifier benchmarks, a candidate program is judged exclusively by executing the supplied test cases. The human “reference solution” included in the original dataset helps design test cases, but it plays no role at evaluation time. Hence reliability is governed by the provided test-case and not the presence of a reference program. Any critique of insufficient “ground truth” therefore applies equally to the original benchmark and lies outside the scope of our transformation procedure.
> > >
> > > **2. Can an incorrect program slip through?**
> > >
> > > Regarding the case of a generated fundamentally incorrect solution passing all test cases in our benchmark, this is not possible in our approach. In Section 2.2, we mention that "We always select the ground truth solution as the solutions which passes all predefined test cases". Consequently, we also apply the ground-truth reference solution from the original to transformed benchmarks as the canonical best solution.
> > >
> > > **3. How many tests are “enough”? Saturation analysis**
> > >
> > > As mentioned above this parallels the question of how many test cases are suitable for the original work because we use the same references and ground truth. To further alleviate this concern, we have conducted saturation plots on HE-R+ and MBPP-R+ where we take the produced benchmarks and simulate how increasing K selected number of test cases effects ranking stability. The results of this study is as follows:
> > >
> > > **HE-R+**
> > > |  k | ρ mean | ρ 95 % CI (low) | ρ 95 % CI (high) | σρ   | Top-1 |
> > > |---:|-------:|---------------:|-----------------:|------:|------:|
> > > |  1 | 0.725 | 0.722 | 0.728 | 0.016 | 1.00 |
> > > |  2 | 0.815 | 0.813 | 0.817 | 0.011 | 1.00 |
> > > |  3 | 0.857 | 0.855 | 0.859 | 0.009 | 1.00 |
> > > |  4 | 0.879 | 0.878 | 0.881 | 0.009 | 1.00 |
> > > |  5 | 0.895 | 0.894 | 0.897 | 0.008 | 1.00 |
> > > |  6 | 0.905 | 0.903 | 0.906 | 0.008 | 1.00 |
> > > |  7 | 0.913 | 0.912 | 0.914 | 0.006 | 1.00 |
> > > |  8 | 0.919 | 0.917 | 0.920 | 0.007 | 1.00 |
> > > |  9 | 0.925 | 0.924 | 0.926 | 0.007 | 1.00 |
> > > | 10 | 0.929 | 0.927 | 0.930 | 0.006 | 1.00 |
> > >
> > > **MBPP-R+**
> > > |  k | ρ mean | ρ 95 % CI (low) | ρ 95 % CI (high) | σρ   | Top-1 |
> > > |---:|-------:|---------------:|-----------------:|-----:|------:|
> > > |  1 | 0.739 | 0.737 | 0.740 | 0.008 | 1.00 |
> > > |  2 | 0.834 | 0.833 | 0.836 | 0.007 | 1.00 |
> > > |  3 | 0.875 | 0.874 | 0.876 | 0.005 | 1.00 |
> > > |  4 | 0.900 | 0.899 | 0.901 | 0.006 | 1.00 |
> > > |  5 | 0.915 | 0.914 | 0.916 | 0.004 | 1.00 |
> > > |  6 | 0.925 | 0.924 | 0.926 | 0.004 | 1.00 |
> > > |  7 | 0.933 | 0.932 | 0.933 | 0.004 | 1.00 |
> > > |  8 | 0.938 | 0.938 | 0.939 | 0.004 | 1.00 |
> > > |  9 | 0.943 | 0.942 | 0.944 | 0.004 | 1.00 |
> > > | 10 | 0.948 | 0.948 | 0.949 | 0.003 | 1.00 |
> > >
> > > The lower bound of the confidence interval exceeds the conventional “high correlation” threshold (ρ ≥ 0.90) at k = 6 for HumanEval+ and k = 5 for MBPP+. Beyond those points the curve flattens sharply. These results show that a modest number of tests already yields stable rankings, consistent with our success in using the limited number of test cases available when transforming the base versions of HumanEval and MBPP (9.6 and 3.0 average number of test cases respectively). We will add full plots and analysis to the camera-ready version of the paper.
> > >
> > > The saturation study provides the requested quality-control check while points 1-2 clarify why the lack of a reference solution is not detrimental. We hope this resolves the outstanding concern.

---

> ### Author Response · Authors · 2025-06-07
>
> Please let us know if there is anything that needs clarification, we would be glad to provide additional information or run quick experiments before the discussion period ends. If our clarifications resolve your concerns, we would greatly appreciate a reconsideration of the current score.
>
> Thank you again for your thoughtful feedback.

---

### Official Review · Reviewer_FFwf · 2025-05-14

**Rating:** 4
**Confidence:** 4
**Ethics Flag:** 1

**Summary:**

This paper proposes a way to transform code generation benchmarks to verification quality benchmarks.

Given some coding problems and tests, they obtain a set of candidate programs for each of the problems and a gold ranking of these candidate programs based on tests. The quality of a verifier is determined by how good the predicted ranking matches the gold ranking.

They used this recipe to transform HumanEval and MBPP to verification benchmarks and evaluate several models' ability to generate tests.

**Questions To Authors:**

Since the authors claim that fine-grained score is important for reinforcement learning, I wonder if there's any concrete evidence that supports this claim. Also, there are several metrics

**Reasons To Accept:**

1. Test generation and their evaluation is an important topic to be discussed.

2. The metrics proposed are multi-facet and cover different aspects of the tests.

3. The techniques and findings are intuitive.

**Reasons To Reject:**

1. Lack of evaluation on more modern coding benchmarks. HumanEval and MBPP are two benchmarks that are largely saturated and too simplistic for modern LLMs. It would be great if you could evaluate the recipe on more modern benchmarks such as NaturalCodeBench, LiveCodeBench, etc.

2. A bit overclaiming about reinforcement learning. On lines 5 and 21, the authors claim that test quality is important for reinforcement learning. While this is an understandable claim, it is not supported by downstream experiments in code-related RL. On line 42, the authors use "reinforcement learning needs a fine-grained score or assessment of a coding solution" to justify their method. Is there any definitive evidence that supports this claim? Because people seem to be using mostly binary rewards these days.

---

> ### Author Response · Authors · 2025-06-01
> **Addressing questions and concerns**
>
> We sincerely appreciate you taking the time to review our paper and address the concerns you mentioned in the review below.
>
> __1. Lack of evaluation on more modern coding benchmarks.__
>
> We agree that our recipe would also apply to more recent coding benchmarks. However, our primary goal was to introduce a general methodology for transforming existing code-generation benchmarks into verification-quality benchmarks, demonstrated using widely accepted datasets like HumanEval, HumanEval+, MBPP and MBPP+. We posit that there is a clear need for this work despite recent reasoning models migrating to benchmarks such as LiveCodeBench. Regarding our chosen benchmarks, these benchmarks are still commonly used and their popularity has decreased mainly in the two months since our paper's submission. At the time of submission, MBPP and HE were standard benchmarks for non-reasoning models and selected in numerous verifier studies from late 2024 and early 2025, laying the foundation for our paper [1, 2, 3, 4, 5]. We also select the non-plus versions of HE and MBPP to limit test our approach in the condition that the number of predefined test cases are low as we describe throughout our work. We believe our work provides a valuable contribution by offering an approach to create verifier benchmarks at a crucial moment when verifier evaluation is becoming increasingly important. We believe it will encourage future works to apply our approach to newer benchmarks such as those you mentioned.
>
> __2. Over claiming the importance of RL.__
>
> We agree with the general sentiment of over claiming RL in our introduction and will soften the phrasing around this as our work focuses on evaluating the verifiers themselves, which are a component in such systems. We were considering works such as [6] which feature fine-grained code rewards but there are certainly other works that feature binary rewards as you mentioned, which our benchmarks are also applicable too. To better reflect the pass/fail rewards in other literature we will change line 42 from “needs” to "may be enhanced by" and encourage the exploration of fine-grained rewards as an interesting concept for RL. We believe our work is also very relevant without the context of RL and will emphasize how such verifiers are important to areas such as filtering SFT data and selecting between parallel compute generations. With regards to our claim in test quality being important for reinforcement learning, our thinking here is that RL’s success is based on the quality of the reward which with predefined test cases is the accuracy of the tests. We think this is justifiable given Deepseek-R1’s success with predefined test cases but can soften the claim to “we suppose” code verification is important for RL.
>
> We are grateful for your thoughts on our work and are eager to hear about any comments you may have!
>
> [1] [https://arxiv.org/abs/2411.05010](https://arxiv.org/abs/2411.05010)
>
> [2] [https://arxiv.org/abs/2502.01715](https://arxiv.org/abs/2502.01715)
>
> [3] [https://arxiv.org/abs/2502.02827](https://arxiv.org/abs/2502.02827)
>
> [4] [https://arxiv.org/abs/2502.01718](https://arxiv.org/abs/2502.01718)
>
> [5] [https://arxiv.org/abs/2411.13611](https://arxiv.org/abs/2411.13611)
>
> [6] [https://arxiv.org/abs/2307.04349](https://arxiv.org/abs/2307.04349)

---

> > ### Comment · Reviewer_FFwf · 2025-06-05
> >
> > Thank you for the response.
> >
> > I'm still not fully convinced of the effectiveness of this recipe, because of the scope of benchmarks selected. Test generation seems relatively easy for these benchmarks but much harder for other ones. Therefore I don't think the findings revealed in the experiments are generalizable, which undermines the claims.

---

> > > ### Author Response · Authors · 2025-06-07
> > >
> > > Please let us know if there is anything that needs clarification, we would be glad to provide additional information or run quick experiments before the discussion period ends. If our clarifications resolve your concerns, we would greatly appreciate a reconsideration of the current score.
> > >
> > > Thank you again for your thoughtful feedback.

---

> ### Author Response · Authors · 2025-06-05
> **Addressing questions and concerns**
>
> Thank you for the feedback! If we understand correctly, the concern is that MBPP and HE are too simple to evaluate test case generation and therefore our recipe may not translate to newer benchmarks like LiveCodeBench for example. There are two components to this critique which we address below:
>
> **1. Transferability of the recipe**
>
> Whether the underlying problems are "easy" or "hard" for code-generating LLMs is not directly a limitation to our approach. Our recipe only needs: (1) problems + reference tests and (2) the ability to sample a diverse pool of candidate solutions. As long as the generator model selected can generate partially correct solutions, we can construct a working benchmark. In the case of LiveCodeBench, O4-Mini achieves 80.2 and DeepSeek-R1-0528 acheives 73.1 Pass@1. This is closely matched by Gpt-4o achieving 72.2 on MBPP+, the generator in our work. Hence the diversity assumption holds and the same sampling-and-ranking pipeline applies unchanged.
>
> **2. Adequacy of MBPP and HumanEval for test-case generation**
>
> Even on these "simpler" sets, test generation is far from saturated as demonstrated in Table 2. When also scaling the number of test cases with top performing reasoning models at the time, the best Top-1 score on MBPP+ we managed to achieve was 81.2 with DeepSeek-R1 and 20 test cases. We also test sufficiently less capable models such as Llama-3.1-8B-Instruct which achieve 55.6 pass@1 on the original MBPP+. Our findings remain consistent when the question difficultly to model capability gap is large, a potential scenario with test case generation applied to more recent benchmarks.
>
> We hope this helps to clarify and address your concerns.

---

### Decision · Program_Chairs · 2025-07-08

**Decision:**

Accept

**Comment:**

The paper presents a pipeline and recipe for transforming code benchmarks into new datasets to evaluate synthetic code verifiers. Reviewers generally acknowledge the need for this work, appreciate the clear presentation, and find the results promising. However, some reviewers raise questions and concerns about the recipe's generalizability and the choice of HumanEval and MBPP as source benchmark datasets. Although the authors provide a comprehensive rebuttal, it does not entirely alleviate the reviewers' concerns. After carefully considering the reviewers' comments and the authors' responses, I believe these concerns are valid but not critical. To address them, the authors should incorporate their responses into the next version of the paper and, preferably, test their framework on additional benchmarks to further demonstrate its effectiveness.